# Women decision making on use of modern family planning methods and associated factors, evidence from PMA Ethiopia

**Fitsum Tariku Fantaye** [1] *, **Solomon Abrha Damtew** [2]

1 FTF Research Consulting PLC, Addis Ababa, Ethiopia, 2 Department of Epidemiology and Biostatistics, Wolaita Sodo University, Wolaita Sodo, Ethiopia

* goodislove1996@gmail.com

**Data Availability Statement:** All relevant data are attached as Supporting Information - Compressed/ZIP File Archive.

**Funding:** The author(s) received no specific funding for this work.

## Abstract

### Background

Family planning decision making is defined as women´s ability to determine the family planning methods that she wanted to use through the process of informed decision making. Despite the availability and accessibility of family planning methods, the utilization rate is not more than 41% in Ethiopia. Evidence and experts have consistently show that women decisions making ability on family planning method they desired to use is one of the possible reasons for this slow rate of family planning use increment. In consideration of this and further motives family planning use decision making has become one of the top sexual and reproductive health related sustainable development agendas. Hence, this study aimed at determining the level, trend and spatial distribution of family planning use decision making among married women and identify factors affecting it.

### Methods

This study was based on Performance Monitoring for Action (PMA) 2020 cross sectional national survey data. Married women who are currently using or recently used family planning method were included in this study. Frequency was computed to describe the study participants while chi-square statistics was computed to examine the overall association of independent variable with family planning use decision making. To identify predictors of family planning use decision making multinomial logistics regression was employed. Results were presented in the form of percentage and relative risk ratio with 95% CI. Candidate variables were selected using p value of 0.25. Significance was declared at p value 0.05.

### Results

This study revealed that one in two women (51.2%; 95% CI: 48.8%-53.6%) decide their family planning use by themselves while 37% (36.8%; 95% CI: 34.5%-39.2%) decide jointly with their husband and/or partner. Women alone family planning use decision making increased significantly 32.8% (95% CI: 29.4%, 36.4%) in 2014 to 51.2% (95% CI: 48.8%, 53.6%) in 2020. It also shows variation across regions from scanty in Afar and Somali to 63.6% in

**Competing interests:** The authors have declared that no competing interests exist.

Amhara region and 61.5 Addis Ababa. Obtaining desired family planning method was found significantly to improve women alone and joint family planning use decision making. Women who have perceive control and feeling if they get pregnant now were found to be positively associated with women alone family planning use decision making. Discussion with husband, his feeling towards family planning were found positively to influence family planning use joint decision making. Moreover, women religion, was found reducing the likelihood of both women alone and joint family planning use decision making while experiencing side effect reduces the likelihood of joint family planning use decision making.

## Conclusion

Half of the women independently decide their family planning use which calls up on further improvement. Family planning use decision making ability is expected to be improved by efforts targeted on husbands' approval on wife's family planning use, discussion on family planning use with husband/partner, improving women psychosociological readiness and trust on her own to decide her desired family planning method; informing the possible side effects and what to do when they encountered during their family planning use visit. In addition, influencing women on the use of family planning via religious leader will help much in this regard. Monitoring and evaluating reproductive health policy 2021 to2025 and addressing bottlenecks which hinder women decision making health service use is hoped to improve women family planning use decision making. Further qualitative study to identify and address factors that contribute for the variation across regions also help much.

## Background

Family planning use decision making is the major component of reproductive health service use empowerment. Though there is no agreed up on definition on family planning use decision making and no single measure [1–5], the following definition is commonly and consistently used across similar studies: Decision making on family planning use is defined as women's ability to independently decide on the family planning method she wanted to use through the process of informed decision making by successfully overcoming unnecessary pressure from significant others around her [6–9].

Until recently, Family planning use decision making has got inadequate attention because the focus of health policies, program and research has been on availability, accessibility and utilization along with determinates of health services as manifested by primary health declaration, save mother initiatives in low- and middle-income countries. To this end, reproductive health services have received global attention in terms of policy articulation, program designing and implementation along with monitoring activities which has guided by researcher [10–22]. Likewise, family planning use decision making got little and/or no attention in Ethiopia.

The Ethiopian government is committed to achieving the Sustainable Development Goal (SDG) of improving maternal health, with a goal of lowering the maternal mortality ratio (MMR) from 401 to 279 per 100,000 and increase contraceptive prevalence rate (CPR) from 41% to 50% by 2025 by promoting reproductive health services, including family planning [22]. The Ethiopia Ministry of Health has also initiated and implemented various programs and activities to make most reproductive health services available and accessible to the community. For instance, the introduction of primary health care with the health extension

program, family planning services provision and community level awareness creation, delivering most reproductive health services free of charge [22, 23], including, maternal and child health services (antenatal care, vaccination, delivery post-natal and family planning services).

Even though immense activities conducted the overall fertility rate and optimal service use, including family planning, could not be altered, or enhanced in most sub-Saharan countries including Ethiopia [24–26]. And also, these efforts did not make optimal service utilization including family planning and the increase in the population size in general, as well as total fertility rate has not been adjusted or reduced.

Recent evidence are showing that women decision making power on reproductive and maternal health services in general and family planning use service in particular is one of the determinantal factor influencing service use, consequently, for the lower rate of contraceptive prevalence rate (CPR) and high fertility [26–34]. This is one reason why most reproductive, maternal, newborn, child health care, service uptake is not optimal despite the availability, accessibility, and affordability of most of those services in Ethiopia also [21, 23, 26, 35–38]. Hence, over the past 5 years, both the global and the national community show a paradigm shift in making family planning service use optimal [39] with a call for addressing such bottlenecks hindering family planning service utilization. This is manifested by the inclusion of women decision-making ability on family planning use as top sub agenda in the women empowerment main agenda, goal 5.6.1 stated [20, 40–42].

This implies the need to look at the level and determinants of family planning decision making which is the bottlenecks which affect services utilization apart from making services accessible and available. Determinates, however, were not explicitly explored in the existing studies and addressing additional potential variables, including group level variables among others is important.

Therefore, this study aimed to determine the level, trend and spatial distribution of independent decision making on current family planning use and identifying factors associated with it by using theory of planned behavior (TPB). Documenting the level, trend and spatial distribution and identifying the factors affecting family planning use decision making helps towards the achievement of the SDG indicator 5.6.1, national CPR improvement and enhancement of women empowerment on sexual and reproductive health services use in general and family planning use in particular by providing actionable evidence for government and partner actors.

## Methods

This study used cross-sectional data from Performance Monitoring for Action Ethiopia (PMA Ethiopia) 2020. The rationality to use PMA data includes that currently, PMA data is the best available recent and real time data even used by the minster. In addition, PMA collects data by resident enumerators using smart phone with customized ODK application which facilitates real time data collection and timely feedback in correcting errors. The hypothesis is tested in this study using a quantitative research methodology by assessing the association between the independent and dependent variables using the theory of planned behavior (TPB) as a guiding conceptual model. And TPB was selected for one thing it specifically focuses on the behavioral aspect of decision making and provides a comprehensive framework for understanding how individuals make decisions about their actions. The theory considers both individual attitudes and subjective norms, as well as perceived behavioral control, which are all important factors in determining an individual's likelihood of engaging in a specific behavior. Additionally, the TPB is a well-established and widely researched theory, making it a reliable and valid tool for predicting and understanding human behavior. Secondly, constructs of this model can be easily derived from variables available in secondary data used in this study.

## Source and study population

The source population for this study were women of reproductive age groups and the study population was restricted to married or cohabiting women who are currently using and/or most recently (in the last two years) used modern family planning (mFP) methods and completed female questioner result were used as inclusion criteria.

## Sample size and selection techniques

A representative sample survey Performance Monitoring for Action (PMA-Ethiopia 2020) was used to provide national level reports.

All women between the ages of 15 and 49 who reside in the chosen households were included in the PMA-Ethiopia survey. A two-stage stratified cluster sampling method used to select enumeration areas. A complete census was conducted in the selected enumeration areas followed by a selection of 35 households per enumeration area using simple random sampling. All reproductive age women were interviewed after the household survey. The PMA-Ethiopia survey offers important data that may be used to track health developments in crucial areas of Ethiopian health system, such as mother and child health including family planning use and decision making, sexual violence, education, service delivery information on family planning service provision and other relevant newborn, maternal and child health data. Six-round survey as PMA 2020 since 2014 was carried out in Ethiopia, followed by expanded cross sectional and panel survey since 2019. It was executed by Addis Ababa University's School of Public Health in collaborative efforts with the Ethiopian Public Health Association with assistance from the Federal Ministry of Health, Central Statistical Agency, The Foreign, Commonwealth & Development Office (FCDO) (formerly DFID), Bill & Melinda Gates Institute for Population and Reproductive Health (Johns Hopkins Bloomberg School of Public Health), JHSPH, Marie Stops International Ethiopia Office (MSI Ethiopia), and the source of funding is from FCDO and the Bill & Melinda Gates Foundation.

The main sample units or enumeration areas (EAs) were chosen using the frame to Ethiopia Population and Housing Census (PHC), which was performed in 2019 by the Ethiopia Central Statistical Agency. As a result, the sample data is neither uniform or randomly distributed, and observations are chosen using a process other than simple random sampling, sometimes referred as complex survey sampling, entails a variety of selection probabilities at multiple stages. The likelihood of selection has an inverse relationship with each person's weight. Instead of using a simple random sample weight, estimate uses sampling weights that are included with the survey data. A total of 213 EAs were chosen in the first stage, with independent selection in each sample stratum and a probability proportional to EA size.

Using a random number generator software, 35 HHs per cluster were chosen at random from the freshly generated household listing in the second round of selection. All females between the ages of 15 and 49 who were either long-term residents of the chosen HH or guests who slept there the night before the survey were eligible to participate in the interview. The protocol of PMA Ethiopia [43] contains all the details on sample design and selection methods.

The Service Delivery Point (SDP) survey, which covers both public and private facilities that service the designated enumeration areas (EAs) for the HH survey, is another component of the PMA Ethiopia. Following the identification of EAs, a list of all public and private health facilities—including all health posts, health centers, and primary level hospitals in associated districts—was collected from the local district health offices. Information regarding the RMNH services that private facilities provide is also gathered, even though private health facilities are still relatively uncommon in Ethiopia. The list of all private health facilities in each

kebele, the lowest level administrative unit in Ethiopia that typically contains five EAs, is examined to sample private health institutions. Within the kebele limits, a maximum of three private SDPs are chosen at random for an interview [43].

For this study, a sample of 2302 married or cohabiting individuals between the ages of 15 and 49 who currently or recently (within two years) used family planning methods and were suitable for our analysis considering our purpose were chosen.

Those women who were not using family planning methods currently or recently, un-married or not in a union, and traditional family planning methods users were excluded from the analysis. Thus, out of the total 7629 reproductive age women included in the PMA 2020, 4880 women who were not using family planning methods during the survey were dropped at initial step. Following the exclusion of 300 women who were not married or living together at the time of the survey, 147 women who used traditional method were also dropped, leaving an unweighted sample size of 2302 women. We used sampling weight to address the disproportionate sample allocation in the PMA, resulting in a final weighted sample size for this study of 2269 (Fig 1).

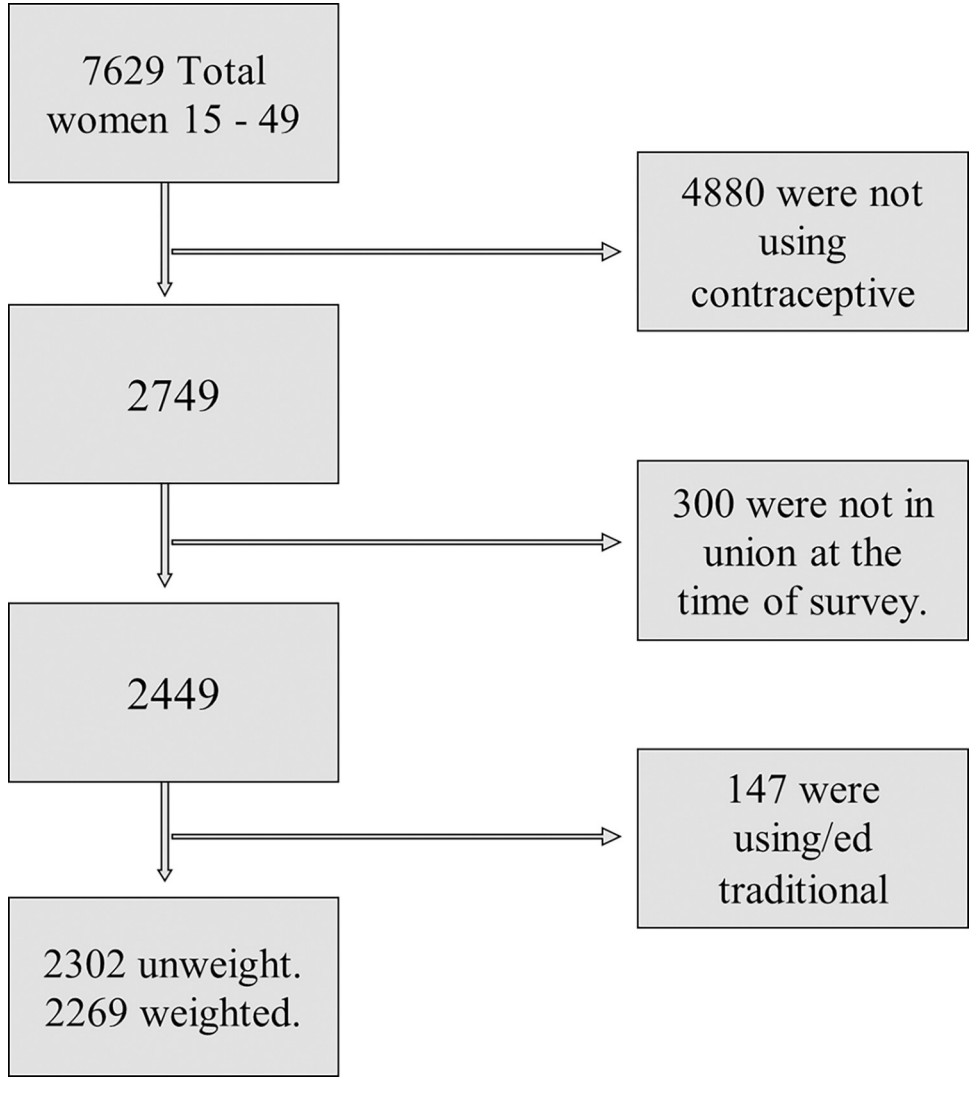

**Fig 1. Schematic presentation of the women included in this study using the 2020 PMA data.**

## Data collection tool and technique

Women's background characteristics (including age, marital status, and education), birth history, knowledge and use of contraception, experience with and perceptions of its side effects, partner contraception feeling and IPV, family planning method use decision, individual and subjective attitude toward contraception, communications with healthcare providers and facility visit were all obtained using the PMA Ethiopia Females Questionnaire. The data gathered by the household questionnaire was also used to determine the GPS location of the households, as well as data on ownership of different durable goods that was used to calculate the wealth index. The service delivery point (SDP) survey questioner was also used to obtain GPS points of the facilities and their type.

## Dependent variable

"Decision on the utilization of current and/or recent FP" were the study's outcome variable. Measuring decision on reproductive and health commodities and services' is generally challenging, since getting standard set of questions is controversial. Scholars often use various metrics since there is no one "gold standard" for evaluating women's empowerment and decision-making.

As shown in Table 1 the dependent variable questioned 'who made the final decision about what method you got?', with six response categories was dichotomized for analysis purposes into "important others = 0" (for married/cohabitated reproductive age women who reported that the decision on their FP use was made mainly by provider, partner, you and provider, and other) and "you alone = 1" (for married/cohabitated reproductive age women who reported that the decision on their family planning method use was made only by themselves). Finally, "you and partner = 2" (for married/cohabitated reproductive age women who reported that the decision on their family planning method use was made by respondent and her husband/ partner jointly).

## Independent variable

Independent variables were classified into individual-level variables and group-level variables broadly. Individual-level independent variables further categorized into socio-demographic characteristics variables, fertility and SRH characteristics variables, family planning method use characteristics variables, husband/partner characteristics variables, socio-phycological variables.

Socio-demographic related variables were respondents age, educational status, religion, and wealth quantile. Parity, marriage history and type, fertility desire, feeling if got pregnant, age at first FP use, number of children at first FP use, were fertility and SRH related variables. Family planning method knowledge, media exposer to FP, obtained desired method, informed side

**Table 1. Description of the dependent variable.**

| FP Decision Making | Variable | Question & Responses | | Categories |
|---|---|---|---|---|
| | | Item | Response | |
| | FPDM | who made final decision on current method | You alone = 1 | 1 = You alone |
| | | | Provider = 2 | 0 = Important others |
| | | | Partner = 3 | |
| | | | You and Provider = 4 | |
| | | | Other = 96 | |
| | | | You and Partner = 5 | 2 = You and partner |

effect was included in family planning method related variables whereas partner discussion before use, partner know FP method used, ipv, husband forced pregnancy, husband FP use feeling were included with husband/partner related variables. Socio-phycological variables which are constructs of TPB include attitude, subjective norm, and perceived control.

Group level variables include community wealth, community education, residence, region, SDP proximity and type. "Region" was grouped into five categories "other regions" include Afar, Somali, Benishangul, and Gambella, Harari, Dire Dawa regions = 0. The remaining regions except Tigray (b/c of the outbreak of the existing war at study period Tigray region were not included in 2020 PMA) were categorized 3 = Amhara, 4 = Oromia, 7 = SNNPRs and 10 = Addis Ababa administrative cite.

"Family planning method knowledge" was generated by sum up responses to the nine-family planning method knowledge questions and further categorizes into two groups if 1 = 'poor knowledge' if respondent heard of 0–4 family planning methods and 2 = 'good knowledge' if respondents heard of 5 to 9 family planning method.

"FP exposure to mass-to-mass media" was formed from the variables (watching tv, listening to radio, and reading the newspaper and social media about FP). As a result, women who watch TV, listen to the radio, or read on social media or newspaper about FP at least once were classified as having exposure to mass media (coded = Yes "1"), whereas those who did not do any of those things were classified as not having exposure to the media (coded = No "0").

"IPV sex physical" was obtained from 'ipv threaten stalk', 'ipv push slap punch kick', 'ipv force pressure sex' variables and further categorized as 0 = 'no ipv' if the respondent didn't experience any sexual or physical violence and 1 = 'at least one ipv' if the respondent experience at least one sexual or physical violence.

"Husband/Part Preg Force" variable was created by sum up three variables and categorized 0 = 'not force' if none and 1 = 'forced pregnancy' if whether the respondents reported that her husband/partner forced her by treated by will have a baby with other women, will leave her, and forced her to get pregnant.

Socio-phycological variables provide assessments of perceived control over FP usage, subjective norms, and attitudes generated from the TPB. These constructs are evaluated using respondents' responses on a five-point Likert scale, where 1 mark "strongly agree" and 5 marks "strongly disagree," in response to various items. As a result of the reversal of these response scales, greater scores now correspond to increased observed pressure. Attitude is constructed by five Likert scale variables, 'fp aut seek partner', 'fp aut trouble prg', 'fp aut could conflict', 'fp aut abnormal birth', 'fp aut side effects disrupt' and sum up and categorize 0 = 'non favorable attitude' if blow mean and 1 = 'favorable attitude' if above mean. Subjective norm also constructed by summing four variables and categorized 0 = 'low subjective nom' if above mean and 1 = 'high subjective nom' if below mean. Four variables summed up to construct Perceived control which is categorized to 0 = 'not able to decide' if below mean and 1 = 'able to decide' if above mean.

## Analysis

Four data sets, namely, household, female respondent, service delivery point and GPS coordinate data sets were used for this study. Before the actual analysis the combined household and female date set was merged with the service delivery point (SDP) data.

Stata v16 was used for this analysis. Frequencies and percentages were computed to characterize the study population. Cross tabulations, chi-square test statistics was computed to see the overall association of the independent variables with the three categories of women decision making.

Frequency was run for every variable to check item nonresponse rate and don't now response which were later excluded from the analysis. Following these variables were recoded to create biological plausible categories. These are followed by checking distribution of the variable using mean and proportion whenever appropriate categories were merged to make cell sample size adequacy. Composite variable was created for variable such the three components of TPB, family planning methods provide, knowledge on family planning methods etc. The nearest service delivery points/type that served the respective enumerations areas (EA) was identified by using the household and service delivery point GPS coordinates.

Multicollinearity was checked and no sign of multicollinearity was detected except two EA level and HH level wealth index variables. The correlation coefficient for these two variables was 0.8602, hence EA level wealth excluded from the analysis.

Moreover, we have tried to check the consistency of constructs for the main predictors of TBA, namely, women attitude, perceive subjective norm and women perceived control and acceptable Cronbach alpha value, ranging from 0.6 to 0.7.

Multinomial logistics regression was used to identify important predictors of women alone and joint family planning use decision making. At bivariate analysis a p value cut of 0.25 was used to select candidate variable for multinomial multivariable logistics regression analysis [44]. Results were presented in the form of percentage, chi-square value and relative risk ratio with 95% CI. Significance was declared at a significance level of 0.05. Except the chi-square analysis all analysis were based weighted count. Svyset command with weight, primary sampling using and stratification to consider the survey sample design.

Four models were run; sociodemographic variable and family planning method related characteristics were entered in the first model; party and fertility and husband related characteristics were entered in the second and third models respectively. Theory of planned behavior related characteristics and proximity and type of SDP variables were entered in the fourth model. Model goodness-of-fit test was checked using the command «mlogitgof» which evaluate the fit of a multinomial logit model to the data and the result shown that the model is good fit to the data meaning that variables included in the final multilevel multinomial logistics regression model explains for the variation in the women decision making on her family planning use. This is supported by the model goodness-of-fit test result; a p value of = 0.969 with a chi-square statistics of 7.119. Only descriptive was used to assess the trend of women decision on their family method use.

Spatial distribution ArcGIS software version 10.4 was also used to visualize the national spatial distribution of level of family planning use decision making by facilities with respect to as the proximity of households by region and cluster. Note that the geospatial distribution does not show data from Tigray.

## Data quality management and control

In PMA Ethiopia survey, data were collected by well experienced PMA filed staff, resident enumerators workers using smart phones Open Data Kit (ODK) system using by real time data generation and timely feedback and questionnaires appeared in three local languages (Amharic, Afan Oromo, and Tigrigna). Weekly error progress report and response, Close follow up during listing, household, and female questioner data collection. 10% reinterview and random checkups by supervisors conducted. Data was checked for completeness and consistency for all completed questioners, those with complete response were considered for analysis.

Even though PMA Ethiopia data have been cleaned, in order to ensure its appropriateness for analysis, data cleaning and quality before conducting different analyses techniques was be employed in this study to exclude the missing values in each variable.

Type of family planning methods include both temporary (Short and long-acting methods) and permanent or irreversible methods.

## Ethical consideration

This study involved a secondary analysis of anonymized data from the PMA Ethiopia. The PMA Ethiopia survey was conducted strictly under the Ethical rules and regulations of world health organization and IIRB of Ethiopian Health and Nutrition Research Institute (EHNRI) and the College of Health Sciences at Addis Ababa University. Informed consent was obtained from respondents during the data collection process of PMA Ethiopia on data collection on oct 2020. The researcher was also obtained formal approval letter to use the data from PMA Ethiopia project and obtained IRB from Addis Ababa university College of Developmental Studies (CoDS) Institutional Review Bord (IRB-CoDS).

## Results

A total of 2269 married/cohabiting women aged 15 to 49 who are currently or recently used family planning method, provided a complete information. Tables 2 and 3 show descriptive results on family planning use decision making and various categories of explanatory variables.

The descriptive result shows that one fourth 26.01 percent of the women were aged between 25–29 whereas 34.07% out of the total women had no education, while 41.29 percent and 24.63 percent reported that they had primary and secondary or higher education respectively. It was realized that women residing in rural areas (68.44%) exceeded their urban counterparts and the majorities have 4 to 5 household members 40.5 percent. Christian orthodox was the leading religion (52.04%) followed by protestant (28.13%) whereas regarding region of residence, 41 out of 100 women were reside in Oromia region and only 16.8% of women were in the lowest quantile (Table 2). One third had more than four children (32.3%) whilst 9.69 percent had no child. And women who were unhappy if they got pregnant by then constitute half (48.7%) but nearly ¾ (71.3%) of women have an intention to have another child. In addition, one-third start sex in the age category 10 to 15 years while 54% start using family planning method in the in-age group 21 to 47 years and 34.25 percent of women had no child at the time of starting family planning use. 79% of the women were reported that they heard and know more than four methods, while about 60 percent had no information of family planning in the last 12 months. Women who reported that they obtained the FP method they desired and informed about the possible side effects of this methods were 88.9% and 25.6% respectively.

The table also show 90.9% and 78.9% of women reported that their husband and/or partner know about the family planning method they used, and discussions made before use respectively. Four percent of the women reported that their husband and/or partner deny their family planning use and 5.3% of them forced to get pregnant. The result also showed that 8% of women had experienced sexual and/or physical violence.

Table 3 showed that the distribution of the socio-psychological variables which include attitude, subjective norm and perceived control based on three constructs of theory of planned behavior (TBP). Women who had favorable attitude towards FP constitute 56% whereas 43.8% had non favorable attitude. Sixty four percent of women had positive subjective norm whereas 35.1 percent had a negative subjective norm towards family planning use. Women who have a perceived ability constitute 73.0% while 26.1% reported they are not able to control. Almost half 46.3% of the women located around 0 to 1 km distance from family planning providing service delivery point.

**Table 2. Distribution of women by selected independent variables, PMA 2020 (weighted, n = 2,269).**

| Variables | Category | Freq. | Percent |
|---|---|---|---|
| | 15–19 years | 140 | 6.18 |
| | 20–24 years | 490 | 21.58 |
| Age | 25–29 years | 590 | 26.01 |
| | 30–34 years | 472 | 20.79 |
| | 35–49 years | 577 | 25.43 |
| | No Education | 773 | 34.07 |
| Education | Primary | 937 | 41.29 |
| | Secondary Plus | 559 | 24.63 |
| Residence | Urban | 716 | 31.56 |
| | Rural | 1553 | 68.44 |
| Regions | Other Regions | 64 | 2.82 |
| | Amhara | 669 | 29.46 |
| | Oromia | 999 | 44.04 |
| | SNNPR | 415 | 18.27 |
| | Addis Ababa | 123 | 5.41 |
| Religion* | Orthodox | 1164 | 52.04 |
| | Protestant | 629 | 28.13 |
| | Muslim | 444 | 19.83 |
| | Others* | 34 | 1.51 |
| wealth quintile | Lowest quintile | 382 | 16.82 |
| | Lower quintile | 443 | 19.52 |
| | Middle quintile | 444 | 19.56 |
| | Higher quintile | 428 | 18.87 |
| | Highest quintile | 572 | 25.23 |
| Marriage history | Only once | 1970 | 86.84 |
| | More than once | 299 | 13.16 |
| Marriage type | Monogamy | 2113 | 93.31 |
| | Polygamy | 151 | 6.69 |
| | No child | 220 | 9.69 |
| 1 child | | 534 | 23.56 |
| Parity | 2 Children | 461 | 20.33 |
| | 3 children | 320 | 14.11 |
| | 4+ children | 733 | 32.32 |
| Fertility Intention (n = 2083) | No More Another child | 598 | 28.7 |
| | Intended Have Another child | 1485 | 71.3 |
| Feeling if got pregnant | Happy | 708 | 34.24 |
| | Mixed Happy | 352 | 17.03 |
| | Unhappy | 1008 | 48.73 |
| Age at first use (n = 2247) | 10 to 20 years | 1018 | 45.31 |
| | 21 to 47 years | 1229 | 54.69 |
| Number of children at first use | No child | 770 | 34.25 |
| | 1 to 2 children | 930 | 41.39 |
| | More than 3 | 547 | 24.36 |
| FP Knowledge | Poor Knowledge | 456 | 20.09 |
| | Good Knowledge | 1813 | 79.91 |
| FP exposure to mass-to-mass media | No media exposure | 1342 | 59.34 |
| | At least one exposure | 919 | 40.66 |

(*Continued*)

**Table 2.** (Continued)

| Variables | Category | Freq. | Percent |
|---|---|---|---|
| Fp obtained desired | No | 251 | 11.08 |
|  | Yes | 2018 | 88.92 |
| Informed FP side effects (n = 2068) | No | 1687 | 74.4 |
|  | Yes | 581 | 25.6 |
| Discussion before use | No | 477 | 21.02 |
|  | Yes | 1792 | 78.98 |
| Partner knows FP use | No | 205 | 9.02 |
|  | Yes | 2064 | 90.98 |
| Physical and/or sexual violence (IPV) (n = 2239) | No IPV | 2060 | 92 |
|  | experience At least one IPV | 179 | 8 |
| Husband forced pregnancy | Not Forced | 2119 | 94.64 |
|  | Forced | 120 | 5.36 |
| Husband FP use Feeling | He disapproves it | 446 | 19.66 |
|  | He is ok with it | 1823 | 80.34 |

As shown in Fig 2 nearly 50% of the women in all the three nearest family planning service delivery distance categories, family planning decision is made by women alone. The tabulation also showed the percentage of married women who use family planning methods and decided by themselves decreased as distance from the SDP increased: 53.08%, 52.5%, and 48.35% for women living less than 1 kilometers, 1 to 2 kilometers, and 2 and more kilometers, respectively (Fig 2).

## Level, trend and special distribution of family planning decision making

**Level of FPDM.** Table 4 and Fig 3 describe the weighted distribution of family planning use decision making. It indicates only for those woman's who are currently married or living in cohabitation with their partner. Table 4 below showed that weighted proportion of level of family planning use decision making with 95% CI. Half of the women's decisions on family

**Table 3. Distribution of women by socio-psychological characteristics and proximity variables, PMA 2020 (weighted, n = 2,269).**

| Variables | Category | Freq. | Percent |
|---|---|---|---|
|  | Non favorable Attitude | 994 | 43.83 |
| Attitude | Favorable Attituded | 1275 | 56.17 |
| Subjective Norm (n = 2267) | Negative Subjective norm | 797 | 35.14 |
|  | Positive Subjective norm | 1470 | 64.86 |
| Perceive Control | Not Able to Decide | 594 | 26.16 |
|  | Able to Decide | 1675 | 73.84 |
| Km Nearest FP_SDP | 0 to 0.99km | 1051 | 46.32 |
|  | 1 to 1.99 km | 596 | 26.26 |
|  | 2km and above | 622 | 27.42 |
|  | Hospital | 41 | 1.80 |
| Nearest FP SDP Type | Health center | 548 | 24.13 |
|  | Health post | 1308 | 57.65 |
|  | Health clinic | 372 | 16.41 |

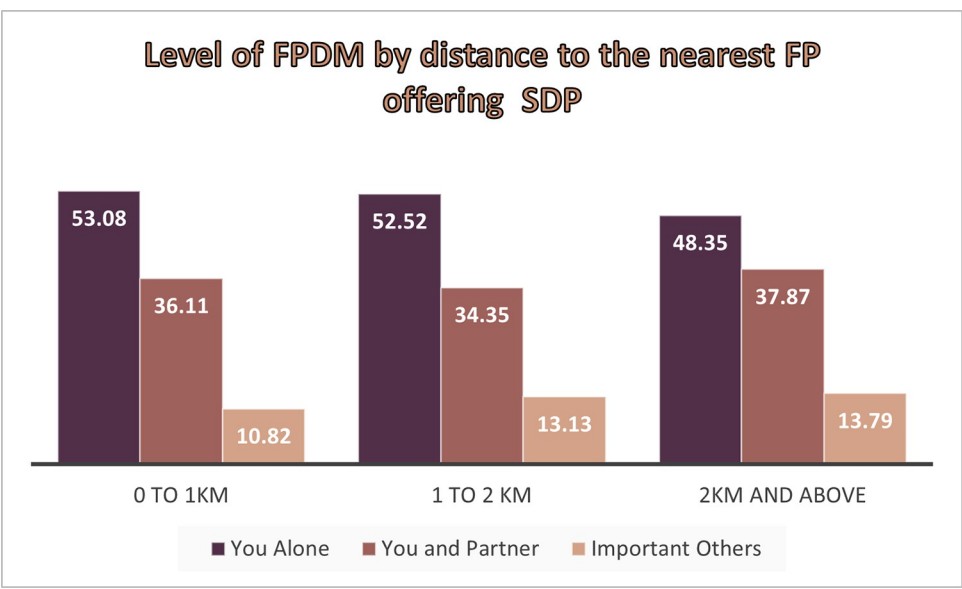

**Fig 2. Family planning use decision making by the nearest family planning providing facilities.**

planning use by themselves alone 51.22% (95% CI; 48.8% - 53.6%), whereas joint decision making was found to be 36.8% (95% CI; 34.5% - 39.2%) (Fig 3).

**Trend of family planning use decision making.** As shown in the Fig 4, the level of FPDM showed statistically significant change from 2014 to 2020. Accordingly, proportion of women who decided by themselves increased 32.8% (95% CI: 29.4%, 36.4%) in 2014 to 51.2% 95% CI: 48.8%, 53.6%) in 2020. Over the same period, joint family planning use decision making decreased from 49.4% (95% CI: 45.6%, 53.1%) to 36.8% 95% CI: 34.5%, 39.2%). Moreover, the proportion of women whose current and/or recent family planning method used was decided by important others show decreased from 17.8% (95% CI: 14.9%, 21.2%) to 12.0% 95% CI: 10.5%, 13.6%) Over the same period of time (Fig 4).

**Spatial distribution of family planning use decision making.** In Ethiopia, a substantial variation in Family planning use decision making is observed across regions. It negligible in the Ethiopian Somali and Afar. Specifically, women alone family planning use decision making ranges from 44.2% in southern nations, nationalities, and people's region to 63.6% in Amhara. As shown in Fig 5, there is marked variation in the level family planning decision making: proportion of married women who decide their family planning use by themselves was found 63.6% in the Amhara region followed by Addis Ababa (61.5%) (Fig 5).

Fig 6 showed the percentage of married women family planning decision making with the nearest family planning provided service delivery type. For women whose nearest service

**Table 4. Proportion of women on family planning decision making before categorization (weighted n = 2269).**

| Dependent Variable | Freq. (W) | Proportion | [95%_Conf | Interval] |
|---|---|---|---|---|
| You alone | 1162 | 0.512 | 0.488 | 0.536 |
| Provider | 95 | 0.042 | 0.033 | 0.053 |
| Partner | 59 | 0.026 | 0.020 | 0.034 |
| You and provider | 117 | 0.052 | 0.042 | 0.063 |
| You and partner | 835 | 0.368 | 0.345 | 0.392 |
| Other | 1 | 0.0005 | 0.0007 | 0.0033 |

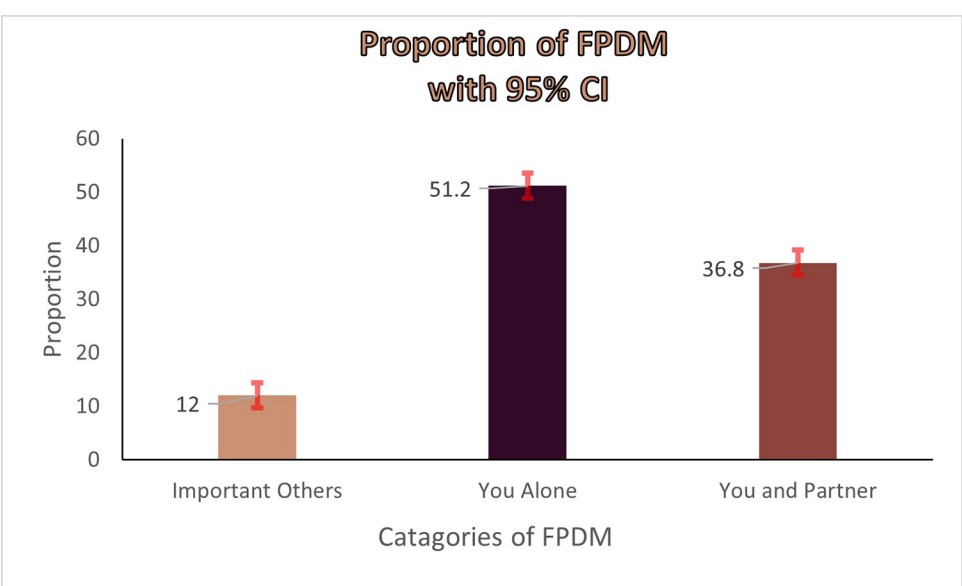

**Fig 3. Proportion of family planning use decision making with 95% CI.**

delivery point was health center, 56.65% of them decided their family planning use by their own, while 44.8% of women whose nearest family planning service delivery was hospital decided their family planning use independently. Strikingly, women decide their family planning use jointly was found to be higher (46.73%) among women who were close to hospital. A little higher than 1/3 of women were found to decide jointly for those women who were closer to health posts (37.22%) and health clinic (37.92) (Fig 6).

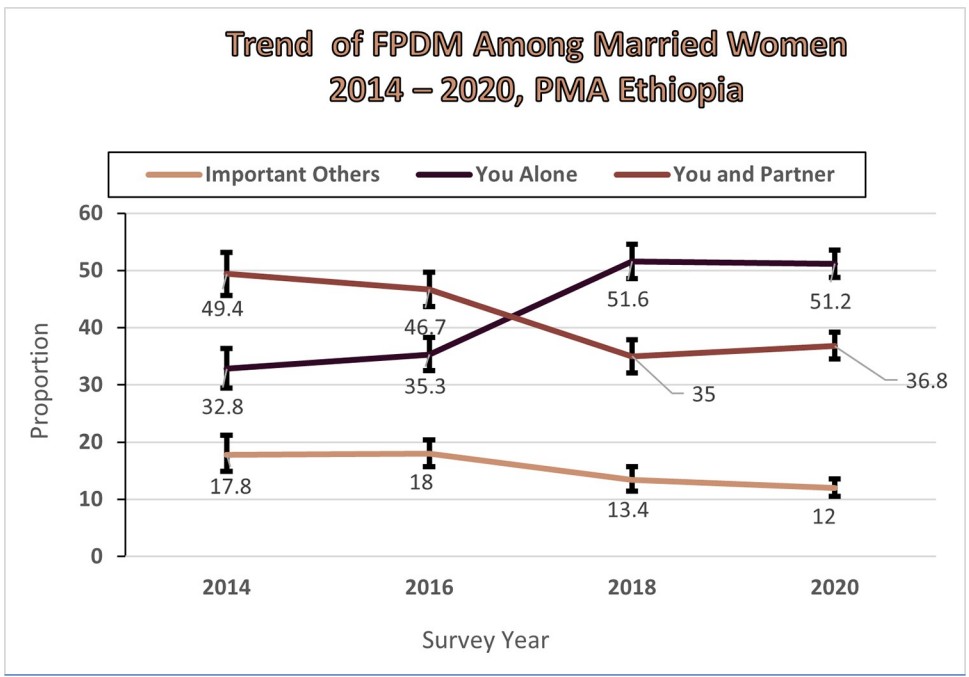

**Fig 4. Trend of family planning use decision making, 2014–2020.**

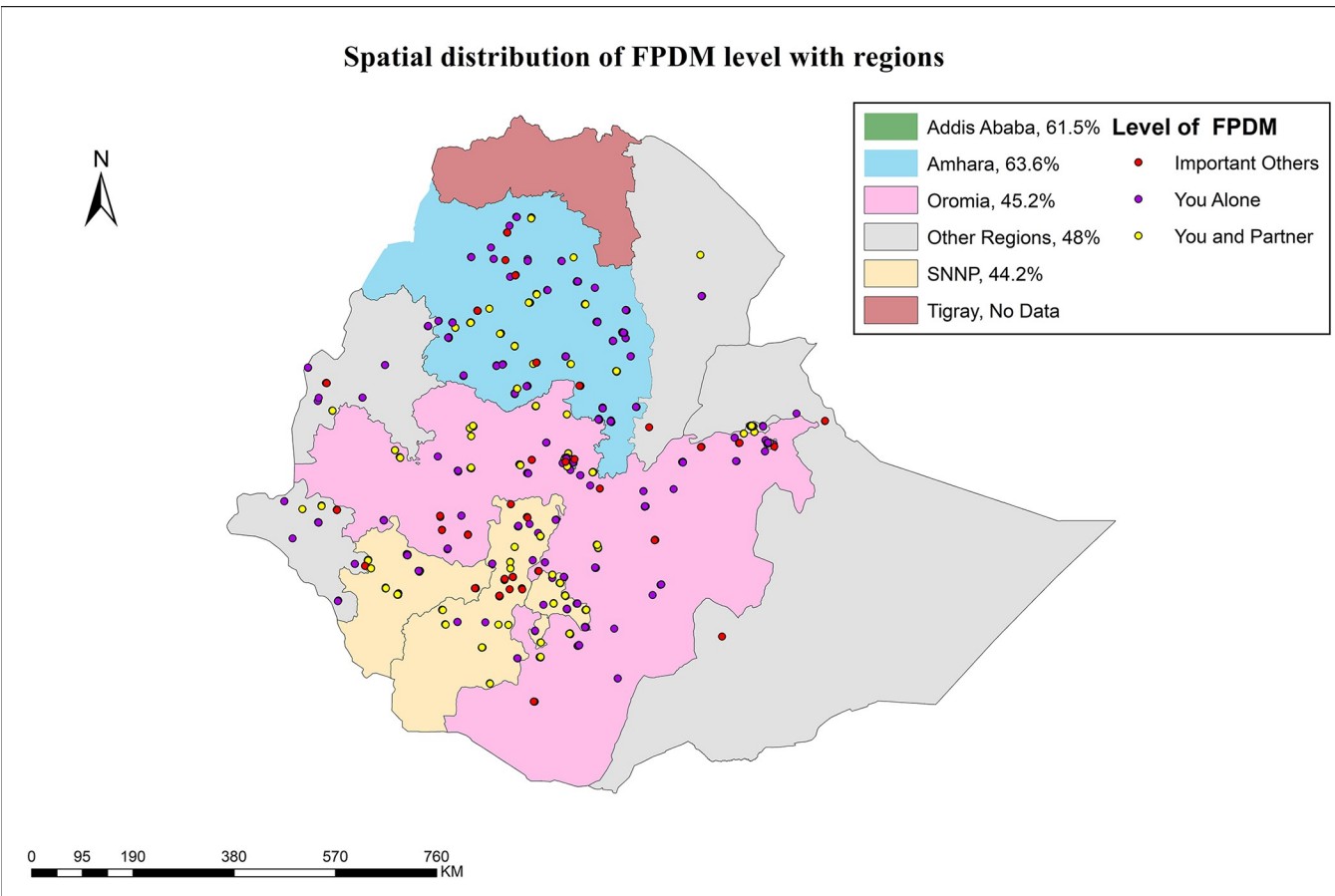

**Fig 5. Geospatial distribution of the level of FPDM by regions.**

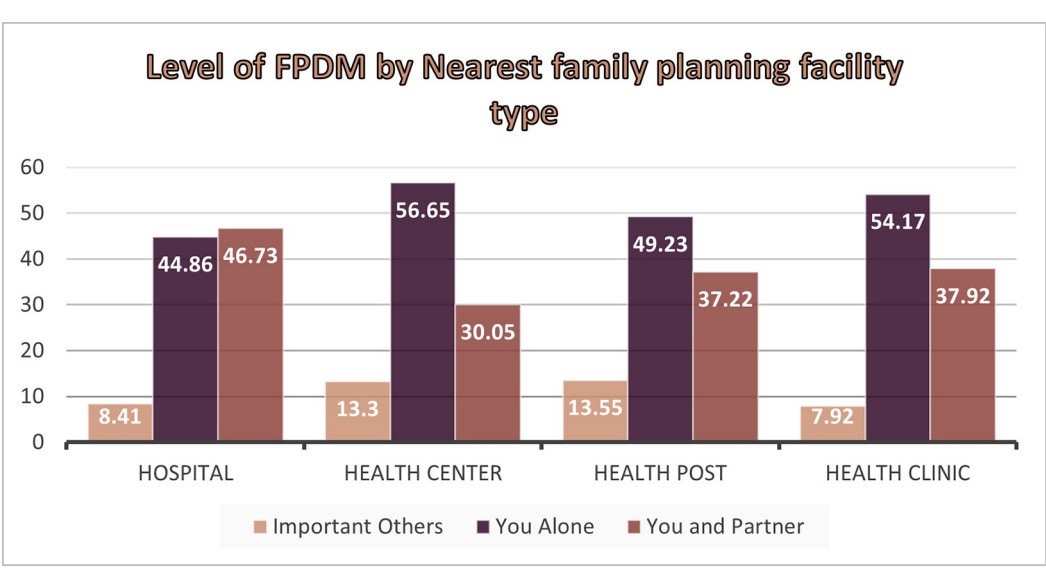

**Fig 6. Level of FPDM by nearest family planning facility type.**

**Table 5. Association between independent variables and family planning use decision making in Ethiopia, 2020 (unweighted, n = 2302).**

| Variables | Categories | FP Decision Making | | | Chi square | P-value |
|---|---|---|---|---|---|---|
| | | Important Others (%) | You Alone (%) | You and Partner (%) | | |
| Religion | Orthodox | 10.37 | 57.58 | 32.05 | 57.06 | 0.00001 |
| | Protestant | 11.76 | 41.49 | 46.74 | | |
| | Muslim | 15.82 | 52.95 | 31.22 | | |
| Regions | Other Regions | 13.53 | 48.07 | 38.41 | 66.58 | 0.00006 |
| | Amhara | 9.55 | 63.65 | 26.8 | | |
| | Oromia | 12.48 | 45.26 | 42.26 | | |
| | SNNPR | 14.26 | 44.23 | 41.51 | | |
| | Addis Ababa | 10.33 | 61.5 | 28.17 | | |
| Feeling if got pregnant | Happy | 13.79 | 49.34 | 36.87 | 4.66 | 0.3242 |
| | Mixed Happy Unhappy | 9.89 | 54.28 | 35.83 | | |
| | Unhappy | 12.10 | 52.16 | 35.77 | | |
| Number of children at first use | No child | 8.29 | 56.93 | 34.78 | 33.81 | 0.0001 |
| | 1 to 2 children | 12.03 | 49.58 | 38.39 | | |
| | 3 to 12 Children | 18.02 | 47.29 | 34.69 | | |
| Fp obtained desired | No | 37.6 | 35.54 | 26.86 | 166.16 | 0 |
| | Yes | 9.08 | 53.74 | 37.18 | | |
| Informed FP side effects | No | 11.13 | 52.53 | 36.34 | 5.56 | 0.0619 |
| | Yes | 14.76 | 49.92 | 35.32 | | |
| discussion before Use | No | 13.59 | 81.29 | 5.12 | 242.78 | 0.0001 |
| | Yes | 11.71 | 44.68 | 43.6 | | |
| IPV sexul phycl | No IPV | 11.75 | 50.81 | 37.44 | 21.17 | 0.00001 |
| | At least one IPV | 15.73 | 64.04 | 20.22 | | |
| FP Feeling | He disapproved it | 16.71 | 71.93 | 11.37 | 140.59 | 0.00001 |
| | He OK with it | 11.01 | 47.19 | 41.8 | | |
| Perceive Control | Not Able to Decide | 15.81 | 47.72 | 36.47 | 13.64 | 0.0011 |
| | Able to Decide | 10.58) | 53.47 | 35.95 | | |

**Association of independent variables with the level of family planning use decision making.** Table 5 show the cross-tabulation analysis between the selected socio-demographic/economic characteristics, fertility and SRH characteristics, husband or partner characteristics, family planning method use related characteristics, Proximity to SDP and method related characteristics, socio-psychological characteristics variables with family planning use decision making. Pearson chi-square test was computed to evaluate the association between each of the independent variables and the dependent variable i.e. family planning use decision making, hence, it showed the preliminary analysis results on association.

As shown in Table 5 above there is an association between region and family planning decision making were the highest in Amhara region 61.5 percent whereas the lowest at SNNPR 44.23 (chi square = 66.58 and p-value = 0.000) and also future fertility intention (chi square = 8.96 and p-value = 0.011), showed relatively moderate association with FPDM, whereas, number of children at first use showed higher relationship with (chi square = 32.13 and p-value = 0.0008 and chi square = 33.81 and p-value = 0.000) respectively.

The association of feeling if got pregnant and family planning joint decision making is almost equal or closer result for the three categories happy, mixed, unhappy, with 36.87%, 35.83% and 35.77% respectively (chi square = 4.66 and p-value = 0.3242). Moreover, FP obtained desired method found significantly associated with family planning use decision making (chi square = 166.1 and p-value = 0.000). FP side effects on the other hand, showed

weak association with family planning decision making with association (chi square = 5.56 and p-value = 0.061).

As shown in Table 4 above with a p-value = 0.000 there is observed a coherent association between husband/partner characteristics related variables (discussion before use, IPV Sexual/physical, and FP feeling) and family planning decision making. socio-psychological variables perceive control showed an association with family planning decision making (chi square = 13.64 and p-value = 0.001).

**Factors associated with Family planning use decision making (FPDM) among Ethiopian women.** Number children at first use and discussion before used were found to be inversely and statistically associated with women alone Family planning use decision making (FPDM) (Table 6). Accordingly, the likelihood of family planning use decision making by women alone among women who had 1–2 child/ren at first use was found to be 0.54 (RRR; 95% CI; 0.33–0.89) less compared with those who didn't have child at first use while the likelihood for those with 3–12 children was found to be only 0.42 (RRR; 95% CI; 0.15–0.93). Discussion on family planning use before they started using the method reduce the likelihood of family planning use

**Table 6. Multinomial logistic regression analysis of the variables associated with family planning decision making among married Ethiopian women PMA Ethiopia, 2020 CS survey.**

| Variables | Categories | You Alone RRR with 95%c Conf Interval | | You and Partner RRR & 95%c Conf Interval | |
|---|---|---|---|---|---|
| Important Others (base outcome) | | You Alone (C RRR) | You Alone (A RRR) | You and Partner (C RRR) | You and Partner (A RRR) |
| Region | Other Regions | 1 | 1 | - | - |
| | Amhara | 2.23 (1.15–4.32)** | 2.351(1.142–4.84)** | | |
| | Oromia | 1.17 (0.599–2.33) | 0.966 (0.432–2.162) | | |
| | SNNPR | 1.03 (0.50–2.11) | 0.741 (0.318–1.73) | | |
| | Addis Ababa | 2.00 (0.98–4.10) * | 0.98 (0.3773–2.557) | | |
| Religion | Orthodox | - | - | 1 | 1 |
| | Protestant | | | 1.614 (0.829- -3.144) | 1.661 (0.741–3.725) |
| | Muslim | | | 0.457 (0.249–0.837) ** | 0.387 (0.184–0.812)** |
| Feeling if got pregnant | Happy | 1 | 1 | - | - |
| | Mixed Happy and Unhappy | 1.536 (0.915–2.579) | 1.557 (0.872–2.78) | | |
| | Unhappy | 1.265 (0.867–1.845) | 1.76 (1.121–2.79) ** | | |
| Number of children at first use | No child | 1 | 1 | - | - |
| | 1 to 2 children | 0.564 (0.401–0.791)*** | 0.544 (0.332–0.892)** | | |
| | 3 to 12 Children | 0.402 (0.258–0.627)*** | 0.366 (0.145–0.926)** | | |
| FP obtained desired | No | 1 | 1 | 1 | 1 |
| | Yes | 7.621 (4.712–12.327) *** | 9.969 (5.953–16.694)*** | 5.995 (3.098–11.601) *** | 5.785 (2.929–11.426)*** |
| Discussion before use | No | 1 | 1 | 1 | 1 |
| | Yes | 0.609 (0.352–1.055)* | 0.344 (0.184–0.642)*** | 10.363 (4.249–25.274)*** | 6.199 (2.713–14.164)*** |
| Perceive Control | Not Able to Decide | 1 | 1 | - | - |
| | Ablet o Decide | 1.533 (1.061–2.214)** | 1.728 (1.13–2.641)** | | |
| Informed FP side effects | No | - | - | 1 | 1 |
| | Yes | | | 0.793 (0.532–1.182) | 0.545 (0.33–0.90)** |
| Experience violence | No IPV | - | - | 1 | 1 |
| | At least one IPV | | | 037 (0.2–0.70)*** | 0.453 (0.208–0.986)** |
| Husband FP Feeling | He disapp_Does not care | - | - | 1 | 1 |
| | He is OK with it | | | 5.579 (3.593–8.664)*** | 2.701 (1.647–4.429)*** |

*** p < .01 and

** p < .05

decision making by the women alone by 65%, 0.34 (RRR; 95% CI; 0.18–0.64). Unlike women alone family planning use decision making; discussion on family planning use with her husband and/or partner before using the method was found to increase the likelihood for joint family planning use decision making, by 6.20 (RRR; 95% CI; 2.73–14.16) compared with their counter parts.

On the contrary, region, women reaction and/or feeling if they get pregnant by then, obtaining the method she desired, and women perceived control were found to be statistically significant factors which positively influence women independent family planning use decision making (Table 6). The likelihood of family planning decision by the women alone and joint decision making were found to be 9.97 (RRR; 95% CI;5.95–16.69) and 5.79 (RRR; 95% CI; 2.93–11.43) times higher among those who obtained the method they desired compared with those who did not respectively. Among the three constructs of theory of planned behavior perceived control was found to improve the likelihood women alone family planning use decision making. Accordingly, women who have perceived control had 1.73 (RRR; 95% CI; 1.13–2.64) higher likelihood of independent family planning use decision making (FPDM) compared with their counterparts. Another factor found to be significant, namely, women reaction and/or feeling if get pregnant by then was found to be positively and significantly affecting family planning use decision making by the women alone. Accordingly, those women who felt unhappy (unhappy and a sort of unhappy) if they get pregnant by then were 1.76 (RRR; 95% CI; 1.12–2.79) more likely to decide on family planning use by themselves compared with women who reported happy (happy and/or a sort of happy) when asked what they thought of if they get pregnant by then. Similarly, women who were reside in Amhara region 2.35 (RRR; 95% CI 1.14–4.84) times more likely to independently decide their family planning use compared with residents of other regions.

Concerning joint decision on family planning women who follow Muslim religion, being informed about method side effect, experiencing violence were found to reduce likelihood of making decision with their husband and/or partner. While, obtaining the desired family planning method, discussion before FP used, and her husband´s feeling to family planning use were found to be statistically significant factors which positively influence women joint family planning use decision making (Table 6).

Accordingly, Women with Muslim religion reduce the likelihood of joint family planning use decision making by 61%, 0.37 (RRR; 95% CI; 0.18–0.81). Similarly, women who didn't informed about family planning method related side effect were 0.55 (RRR; 95% CI; 0.33–0.90) less likely to decide family planning use jointly. Women who were experiencing at least one form of physical and/or sexual violence were found to have lower 0.45 (RRR; 95% CI; 0.21–0.98) likelihood of joint family planning use decision making compare with those who did not experience. Finally, those women whose husband and/or partner had positive reaction for their family planning use, i.e., was found to be ok on their family planning use have 2.70 (RRR; 95% CI; 1.65–4.43) more likelihood in deciding jointly on their family planning use.

## Discussion

Family planning use decision making is one of the indicators of sustainable development goal (SDG) agendas, hence, documenting the magnitude of women family planning use decision making level and identifying the factors affecting in greater depth has paramount importance in monitoring the achievement of such burning global agenda. Unfortunately, studies conducted in Ethiopia to measure family planning use decision making are limited in number and geographical scope. Therefore, this study used nationally representative update data to

determine the magnitude of women family planning use decision making at national level and identify factors affecting it in much more detail using a more advance analysis technique.

Accordingly, nationally half of women were found to be able to decide family planning use by themselves and a bit more than one third (37%) of women decide with their husband and/or partner. A set of factors which both positively and negatively influence women alone and decision with their husband and/or partner on family planning use were identified.

The level of women alone decision making was in line with other studies 53.8 [3] and 52% [4]. It's found higher than similar study 22% [45], 21.6% [46] 14.2% [1], 1/3 [4]. This might be related with outcome variable measurement difference [1, 4] and time difference and variation in the categories of the output variable where this study includes two additional categories (decision by health care provider alone and deicide jointly with the health care provides) [45, 46].

On the other hand, In this study family planning use decision making with her husband and/or her partner was found to be 37% which was lower than findings from 67% [2, 5], and a study conducted in South Africa 45% [45]. The socioeconomic disparities across the nations might be the cause of the discrepancy; for instance, in South Africa, most women make joint decisions. And also, it might be due to outcome variable measurement, i.e. this study measure decision making by a single question while the two studies use composite variable to measure women decision making on family planning use.

The level of women alone decision making on their family planning use showed nearly a 20% incremental change over 7 years which is accompanied by a significant decrease in the joint decision and decision by important others which is a positive trend that aligns with both national and international indicators, targets, goals, and strategies related to reproductive health and family planning decision-making. Internationally Ethiopia, along with other countries, has committed to achieving the SDGs, including SDG target 5.6.1 which specifically aims to enhance the proportion of women who make their own decision on sexual relations, use of contraceptive and health care by 2030 [40, 41]. Additionally, Ethiopia is a partner country of the global Family Planning 2020 (FP2020) initiative which aims to enable 120 million more women and girls to access voluntary family planning by 2020. One of the core principles of FP2020 is ensuring that individuals have the right to make informed decisions about their sexual and reproductive health, including family planning use [47, 48].

Nationally Ethiopia has set its own indicators and targets for reproductive health and family planning. While specific figures on women's family planning decision-making not available as a separate indicator, it is closely linked to broader target of increase women's RH matters decision making power to 100% by 2025 and increasing contraceptive prevalence rate (CPR) of 55% and reducing unmet need for family planning to 10% by 2025 as part of its RH Strategic Plane and Health Sector Transformation Plan II [22, 42].

Family planning use decision making showed substantial variation across regions in Ethiopia: from very minimal in Afar and Ethiopia Somali to 63.6 in Amhara region and 61.5% in Addis Ababa and this is in line with [49]. This might be related with absence of regional-specific programs and interventions that promote women's empowerment, lack of Improving healthcare facilities and provider training in regions, women and their husband and/or partner´s perception towards family planning methods, external factors such as community attitude and religion might contribute their share as indicated in the Ethiopian RH strategic plane 2020–2025 [42].

In line with studies [28, 31, 32, 50] husband and/or partner related characteristics such as discussion on family planning use before they started to use was found negatively related with women alone decision making while in line with [31, 45] it was found positively related with decision making with husband and/or partner reflecting male dominant decision.

In the multivariate multinomial logistics regression analysis, among the theory of planned behavior constructs perceived control was found to improve women alone family planning use decision making after controlling potential confounders. This is line with [3, 28, 31, 32, 50]. Such finding is interesting in the midst of male dominant decision making in matters pertaining to household level decision making in general and health care service use decision making in particular [51].

In line with studies [52] obtaining the method they desired positively and statistically improved both women alone family planning use decision making and decision with her husband and/or partner. Similarly, feeling unhappy if they got pregnant by then increases its likelihood. Unlike one study on EDHS [46] living in Amhara region was found to increase the likelihood of women alone decision making compared with residents of other regions which is also in line with [49].

On the other hand, number of children at first use and discussion with husband before they used were found to reduce the likelihood of women alone family planning use decision making. In line with studies [28, 31] having more children at first family planning use reduces the likelihood of women alone family planning use decision making. Those women who discussed with their husband and/or partner before they used the method have reduced their chance of family planning decision making by themselves [5, 28, 51]. The finding from qualitative studies [28, 31, 32, 50] perceived control was one of the factors that positively influence women alone family planning use decision making (FPDM)

In line with studies [5, 28, 51], husband and/partner related characteristics namely, discussed with her before she used and his positive reaction on her family planning use were found to increase the likelihood of decision making on family planning use with her husband and/or her partner jointly. Similarly with other study [46] women with Muslim religion reduce the risk of family planning use decision making with her husband and/or her partner jointly which is also unlike the study conducted in South Africa which founds women's with a religion of Methodist, Pentecostal, Seventh Day Adventist (SDA) were less likely to take part in decision-making to use contraception [45]. Experiencing one form physical and/or sexual violence were found reducing the likelihood for decision making with her husband and/or her partner [28, 31, 32, 51].

In contrary to studies age, educational status, residence, [2, 4, 30, 46] wealth quintile, women, subjective norm, attitude, parity, future fertility desire, women knowledge on family planning methods [1, 3, 4, 9], women place residence [5] were not found to be associated with either women alone and/or joint family planning use decision making. This might be due to the difference in sample design, scope of the study and how the outcome variables are measured (some used composite variables and some single variables). The other possible likely reason is the inclusion of more confounders in this study unlike those cited here in and one study focus on knowledge, attitude and overall awareness related factors that affect women family planning use decision making [3]. Similarly, unlike studies [4, 46, 49] exposure to media is even not a candidate variable for multivariate with p value of 0.7 at bivariate. Additionally, unlike studies [53, 54] distance from the health facility were not significant in this study. Actually, the mentioned study examines the effect of distance on family planning use while this study attempted to measure family planning use decision making. if distance affects family planning utilization it is likely to affect family planning use decision making by implication. This might be due to the very nature of the Ethiopian health care tire system, i.e., health facilities at the primary heal care unit mainly health posts very closed to the communities which is also evidenced in this study. In addition, health clinics in urban setting and rural drug vendors along with lower clinics in rural areas are very near to the community which women use as an alternative family palling method sources particularity on market days.

This study is not spared from limitations. To begin with, potential limitation of this study is the fact that we used measures of map distance, which are straight line distances between households and the nearest health facilities, and therefore do not consider access to roads and travel time. However, in the absence of such complete road network information using the maps distances, might be practical. In addition, the trend is measured based on cross sectional data and may not show the clear picture, i.e. survey year variable was not included in the model as a factor since most variables were measured in the most recent cohort surveys. Moreover, because of PMA 2020 didn't collected information on variables such as husband desired number of children and timing of additional child, age difference, husband education and employment, women employment were not measured in this study. Furthermore, as a matter of fact, PMA Ethiopia 2020 cross-sectional survey did not collected data from Tigray region due to the conflict, therefor any form generalization need to consider this in mind. For example, in the geospatial distribution the region is depicted with no data.

As a strength this study addresses potential confounder variables, notable group level variable. This study includes additional variables individual and community level variables including health facility related variables rarely measured variables such violence. Moreover, this was guided by theory of planned behavior (TPB), there has been a lack of such similar studies incorporating TPB.

## Conclusion

One in two women who are using and/or used family planning method were able to decide independently to use the method which calls further improvement to escalate up and maintain near to 100%. Addressing the performance targets set by the 2020–2025 Ethiopian RH Strategic Policy, setting up additional new polices, strategies and interventions that are designed and target to enable every woman hoped to help women fully decide by themselves.

Women alone family planning use decision making increased significantly from 2014 to 2020. This 20% change in the level of women alone decision making is promising and in line with national and international indicators, targets, and strategies. However, it is important to continue monitoring progress, addressing barriers, and ensuring sustained efforts to achieve the desired goals.

The findings of substantial regional disparities in women's alone contraception decision-making highlight the need for targeted interventions in regions with lower rates. Some potential strategies to address these disparities could include setting up regional-specific intervention such as tailoring family planning programs to address the unique cultural and social factors in each region, ensuring that women have access to accurate information and resources to address misconceptions, reduce stigma, and emphasizing the importance of women's participation in decision-making.

This study identified factors which is positively and negatively related with women alone and joint decision making on family planning use. Activities and efforts that help women to be able to obtain the method she desired to use, improve her physiological readiness and trust on her own to use family planning methods contribute their share in improving women alone decision making. Additionally, the findings calls up on creating awareness on reproductive health service availability and accessibility including maximizing use of youth friendly service unit thereby empowering young/adolescent's and girls to use family planning method.

Activities and intervention that improve husband and/or partner involvement in his wife´s family planning use decision making process such as improving his feeling on his wife family planning use, balanced discussion on family planning use where the women say is equally entertained before using the method and encourage women to decide to use family planning

via religious leaders mainly Muslim religious leaders, informing the possible side effects and what to do if the women encounter serious side effect during her family planning use visits are hoped to improve women alone and joint family planning use decision making by the couples.

Finally including missing variable such as husband desired number of children, age difference, husband education and employment, women employment for future research in general and performance monitoring for action (PMA) survey in particular.

## Supporting information

**S1 Data.**
(ZIP)

## Acknowledgments

The authors would like to thank the PMA project for allowing and providing the data for further analyses. And also, our thankfulness also goes to all PMA Ethiopia project data collection teams and study participants.

## Author Contributions

**Conceptualization:** Fitsum Tariku Fantaye, Solomon Abrha Damtew.

**Data curation:** Fitsum Tariku Fantaye, Solomon Abrha Damtew.

**Formal analysis:** Fitsum Tariku Fantaye, Solomon Abrha Damtew.

**Investigation:** Fitsum Tariku Fantaye.

**Methodology:** Fitsum Tariku Fantaye, Solomon Abrha Damtew.

**Project administration:** Solomon Abrha Damtew.

**Software:** Fitsum Tariku Fantaye, Solomon Abrha Damtew.

**Supervision:** Solomon Abrha Damtew.

**Validation:** Fitsum Tariku Fantaye, Solomon Abrha Damtew.

**Visualization:** Fitsum Tariku Fantaye, Solomon Abrha Damtew.

**Writing – original draft:** Fitsum Tariku Fantaye, Solomon Abrha Damtew.

**Writing – review & editing:** Fitsum Tariku Fantaye, Solomon Abrha Damtew.

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
