## [Decision Letter · Decision Letter 0]

8 Aug 2023

PONE-D-23-04330Women Decision Making on Use of Modern Family Planning Methods and Associated Factors, Evidence from PMA EthiopiaPLOS ONE

Dear Mr. Fitsum Tariku Fantaye,

Thank you for submitting your manuscript to PLOS ONE. After careful consideration, we feel that it has merit but does not fully meet PLOS ONE’s publication criteria as it currently stands. Therefore, we invite you to submit a revised version of the manuscript that addresses the points raised during the review process.

We look forward to receiving your revised manuscript.

Kind regards,

Takele Gezahegn Demie, MPH

Academic Editor

PLOS ONE

Journal Requirements:

Additional Editor Comments (if provided):

Dear Author,

Congratulations for your well written manuscripts.

Please address the reviewers' comments in detail and provide a point-by-point response.

I also feel some editorial to be corrected to meet the journal's guideline too.

Reviewers' comments:

Reviewer's Responses to Questions

**Comments to the Author**

1. Is the manuscript technically sound, and do the data support the conclusions?

Reviewer #1: Yes

Reviewer #2: Yes

2. Has the statistical analysis been performed appropriately and rigorously? 

Reviewer #1: Yes

Reviewer #2: Yes

3. Have the authors made all data underlying the findings in their manuscript fully available?

Reviewer #1: Yes

Reviewer #2: No

4. Is the manuscript presented in an intelligible fashion and written in standard English?

Reviewer #1: No

Reviewer #2: Yes

5. Review Comments to the Author

Reviewer #1: 1.make clear your study which type of family planning (long term or temporary family planning)

2 according PMA there are some additional factors involved but not discussed in your study

The methods are generally appropriate, although clarification of a few details and provision of a rationale for the use of this particular method of measuring Women Decision Making on Use of Modern Family Planning Methods and Associated Factors, Evidence from PMA Ethiopia

Reviewer #2: Dear editor thank you for invitation to review the manuscript entitled “Women Decision Making on Use of Modern Family Planning Methods and Associated Factors, Evidence from PMA Ethiopia”. The topic addressed important issue even if it need many things to be corrected

1. From the topic Modern Family Planning Methods has to be replace by Modern contraceptive methods. The two terms contraceptive and family planning are different and we cannot use interchangeably in some context. Also it has to be edited throughout the document accordingly.

Abbreviations has to be avoided from the topic unless it is well known. So PMA has to be written in long form.

2. The word Ethiopia cannot be the key word for this study and the authors has to change it.

3. The authors used TPB is not appropriate for this study. Intension of this study is to examine how women decide to use family planning? But TPB is intended to measure intention to practice some behaviour. The study participants are already practicing the behaviours

4. Source population is not appropriate and has to be reframe according to study topic.

5. Inclusion and exclusion criteria has to be there since data were collected for other purpose and all data might not be appropriate for this study.

6. Appropriate sample which can answer the objective has to be calculated. Or justification has to be there why they have used the current sample size.

7. The chart/graphs used to show the trend has to be line graph. So author have to change Figure 4 or has to draw additional chart.

8. Result, discussion and conclusion were well written.

6. PLOS authors have the option to publish the peer review history of their article (what does this mean?). If published, this will include your full peer review and any attached files.

Reviewer #1: No

Reviewer #2: No

---

## [Author Response · Author response to Decision Letter 0]

6 Sep 2023

Point by point response. 

Reviewer #1: 

Comment 1: make clear your study which type of family planning (long term or temporary family planning)

Response: Both permanent and long acting Contraceptive methods were asked in the study.

Assuming the term long term is to mean permanent method, our study includes permanent methods, notable the contains responses on Female sterilization. 

MCP in this study means it both CM. This is included as at the end of the analysis part in the main manuscript, both versions.

Comment 2: according to PMA there are some additional factors involved but not discussed in your study.

Response: This study tried to exhaustively include as many as 20 variables in the model which are summarized in the descriptive section. Selection of important confounders was based on available empirical evidence in the area of FP use decision making.

The methods are generally appropriate, although clarification of a few details and provision of a rationale for the use of this particular method of measuring Women Decision Making on Use of Modern Family Planning Methods and Associated Factors, Evidence from PMA Ethiopia.

Response: We appreciate this concern, and the following two clarifications are mentioned. To start with, detail clarification about PMA data is also available in the method sample size and selection technique part in the main manuscript. The following paragraph is also added in order to clarify the rationality to use PMA data in the same section. 

The rationality to use PMA data includes that currently, PMA data is the best available recent and real time data even used by the minster. In addition, PMA collects data by resident enumerators using smart phone with customized ODK application which facilitates real time data collection and timely feedback in correcting errors.

If the concern is meant to mean measurement of the outcome variable, family planning decision making, as can be witnessed form the listed studies in the reference and clearly justified in the introduction part ,the absence of standardized measurement is a problem; most studies which include further analysis of Demographic and Health Survey (DHS) data using a single question to measures FP use decision making while few fragmented studies try to used inconsistent constructs to create a composite variable without the evidence on the constructs reliability and reducibility. 

Reviewer #2: 

Comment 1: From the topic Modern Family Planning Methods has to be replace by Modern contraceptive methods. The two terms contraceptive and family planning are different and we cannot use interchangeably in some context. Also it has to be edited throughout the document accordingly. Abbreviations has to be avoided from the topic unless it is well known. So, PMA has to be written in long form.

Response: Comment accepted and correct as per the reviewer comment in the body of the manuscript, except terms referring CPR. 

About the PMA (Performance Monitoring for Action) as abbreviation, the term is known for more than a decade as project name for multicounty studies in Africa and Asia. In addition, elaborating the term makes the title longer, hence, the authors prefer to keep it. Yet if reviewer need to be elaborated, we will correct it in the manuscript production and proof phase.

Comment 2: The word Ethiopia cannot be the key word for this study and the authors has to change it.

Response: Comment accepted the term Ethiopia omitted. 

Comment 3: The authors used TPB is not appropriate for this study. Intension of this study is to examine how women decide to use family planning? But TPB is intended to measure intention to practice some behavior. The study participants are already practicing the behaviors.

Response: The authors acknowledged that TPB is used for intended behavioral practice (intention and developing the specific behavior). The rationality to use this model is also already included in the beginning of the method section. Yet, among the available Models, the authors found TPB better than other’s given our study participants were current users and/or recent users in the past 2 years their intention to use is proximal to their practice. Having acknowledged the valuable comment of the reviewer, we are hoping that guiding our study using the proximal model rather than using simple conceptual framework further improves the manuscript quality.

Comment 4: Source population is not appropriate and has to be reframe according to study topic.

Response: Comment accepted and corrected as per the reviewer’s comment. 

Comment 5: Inclusion and exclusion criteria has to be there since data were collected for other purpose and all data might not be appropriate for this study. 

Response: Comment accepted, and necessary modification made in the study population subsection in the method part in the manuscript.

Comment 6: Appropriate sample which can answer the objective has to be calculated. Or justification has to be there why they have used the current sample size.

Response: We appreciate the reviewer comment, and the authors believed that calculating sample size is part of academic scientific writing. Yet the calculated sample size using single proportion formula applying design effect is 1000, while this secondary data gave a final sample size of 2301, which satisfies an over sample size adequacy.

The next concept as far as sample size is concerned is cell sample adequacy which our data full filled.

Comment 7: The chart/graphs used to show the trend has to be line graph. So, author have to change Figure 4 or has to draw additional chart.

Response: Comment incorporated in the manuscript. We thank you and as you rightly mentioned the use of line graph for time series data is recommended. Tacticians advised its use for repeated cross-sectional data such as PMA, at least with at least 10-time data points is, we have used 4 data points, that is why we preferred to use the bar graph. 

Comment 8: Result, discussion and conclusion were well written.

Response: We are grateful for your positive words on our write up.

---

## [Editor Report · Decision Letter 1]

28 Dec 2023

PONE-D-23-04330R1Women Decision Making on Use of Modern Family Planning Methods and Associated Factors, Evidence from PMA EthiopiaPLOS ONE

Dear Dr. Fantaye,

Thank you for submitting your manuscript to PLOS ONE. After careful consideration, we feel that it has merit but does not fully meet PLOS ONE’s publication criteria as it currently stands. Therefore, we invite you to submit a revised version of the manuscript that addresses the points raised during the review process.

We look forward to receiving your revised manuscript.

Kind regards,

Takele Gezahegn Demie, MPH

Academic Editor

PLOS ONE

Journal Requirements:

Dear Author,

I would like to congratulate you for this well written manuscript with nice methodology and detailed analyses.

I also appreciate your responses and justifications for reviewers' comments/queries!

However, I will add few comments to be addressed before final disposition of the article.

Key words could be written as "Keywords".

Tables: Most of the tables should be edit. Different font styles and size were used. Please check once. Please be consistent!

Figures: Nice to remove the background color (Figure 2 and Figure 6)

Discussion: Discussion on Trend of Family Planning Use Decision Making and Geospatial distribution of the level of FPDM by region are lacking or not sufficient! Please add. Better to substantiate the discussion by adding more description to the interpretation and implication of the main findings!

How your main findings compared to the current national and/or international indicators, targets, or goals as well strategies???

You mentioned "Moreover, this was guided by theory of planned behavior (TPB), no single similar study employed."

I thin this i not good way of expressing lack of other similar study because you might searched such a study and didn't found.

Conclusions and recommendations: I feel it is good to summarize the sections together. Both seems too long. Just conclude for your objectives!

Particularly, it is unusual to add separate section for recommendations and mostly recommendations are summarized along with conclusions.

Even it too long in this manuscript.

As to me, a single paragraph is sufficient! However, some descriptions may go to the discussion section as needed!

Please focus on the main findings and provide brief conclusive statement along with recommendations to together!

I think both should be plural. But, in revision it could be conclusions!

However, that might not be there is no similar study. Still there might be similar study, or studies might underway. So, please revise this statement.

Best wishes!

---

## [Author Response · Author response to Decision Letter 1]

19 Jan 2024

Manuscript Number: PONE-D-23-04330R1

Rebuttal letter

Manuscript Number: PONE-D-23-04330R1

Title: Women Decision Making on Use of Modern Family Planning Methods and Associated Factors, Evidence from PMA Ethiopia. 

Thank you for the opportunity to address the comments from the academic editor and reviewer(s). The authors are very grateful for the constructive and valuable comments provided. After taking the feedback, the authors hope that the Reviewers and Academic Editors will be satisfied with the further amendments we have made to the manuscript.

Point by point response. 

**Detailed comments from Academic Editor**

I would like to congratulate you for this well written manuscript with nice methodology and detailed analyses. I also appreciate your responses and justifications for reviewers' comments/queries!

However, I will add few comments to be addressed before final disposition of the article.

* Key words could be written as "Keywords".

Response: Thank you, and corrected as “Keywords”

* Tables: Most of the tables should be edit. Different font styles and size were used. Please check once. Please be consistent! 

Response: Thank you for identifying this inconsistency and revised. 

* Figures: Nice to remove the background color (Figure 2 and Figure 6)

Response: Thank you for identifying this inconsistency and revised. 

* Discussion: Discussion on Trend of Family Planning Use Decision Making and Geospatial distribution of the level of FPDM by region are lacking or not sufficient! Please add. Better to substantiate the discussion by adding more description to the interpretation and implication of the main findings!

How your main findings compared to the current national and/or international indicators, targets, or goals as well strategies???

Response: To provide additional discussion on the trend and geospatial distribution, the following paragraph was added

For the trend 

“The level of women alone decision making on their family planning use showed nearly a 20 % incremental change over 7 years which is accompanied by a significant decrease in the joint decision and decision by important others which is a positive trend that aligns with both national and international indicators, targets, goals, and strategies related to reproductive health and family planning decision-making. Internationally Ethiopia, along with other countries, has committed to achieving the SDGs, including SDG target 5.6.1 which specifically aims to enhance the proportion of women who make their own decision on sexual relations, use of contraceptive and health care by 2030 [1, 2]. Additionally, Ethiopia is a partner country of the global Family Planning 2020 (FP2020) initiative which aims to enable 120 million more women and girls to access voluntary family planning by 2020. One of the core principles of FP2020 is ensuring that individuals have the right to make informed decisions about their sexual and reproductive health, including family planning use[3, 4]. 

Nationally Ethiopia has set its own indicators and targets for reproductive health and family planning. While specific figures on women's family planning decision-making not available as a separate indicator, it is closely linked to broader target of increase women’s RH matters decision making power to 100% by 2025 and increasing contraceptive prevalence rate (CPR) of 55% and reducing unmet need for family planning to 10% by 2025 as part of its RH Strategic Plane and Health Sector Transformation Plan II [5, 6].”

For the regional disparities.

“This might be related with absence of regional-specific programs and interventions that promote women's empowerment, lack of Improving healthcare facilities and provider training in regions, women and their husband and/or partner´s perception towards family planning methods, external factors such as community attitude and religion might contribute their share as indicated in the Ethiopian RH strategic plane 2020 – 2025 [5].”

* You mentioned "Moreover, this was guided by theory of planned behavior (TPB), no single similar study employed."

I think this is not good way of expressing lack of other similar study because you might searched such a study and didn't found. However, that might not be there is no similar study. Still there might be similar study, or studies might underway. So, please revise this statement.

Response: The authors agree and have made the following changes to the manuscript: ‘there has been a lack of such similar studies incorporating TPB’. 

* Conclusions and recommendations: I feel it is good to summarize the sections together. Both seems too long. Just conclude for your objectives!

Particularly, it is unusual to add separate section for recommendations and mostly recommendations are summarized along with conclusions.

Even it too long in this manuscript. As to me, a single paragraph is sufficient! However, some descriptions may go to the discussion section as needed!

Please focus on the main findings and provide brief conclusive statement along with recommendations to together!

I think both should be plural. But, in revision it could be conclusions!

Response: The authors agree and have made the following summarized changes to the conclusion section: 

“One in two women who are using and/or used family planning method were able to decide independently to use the method which calls further improvement to escalate up and maintain near to 100%. Addressing the performance targets set by the 2020 – 2025 Ethiopian RH Strategic Policy, setting up additional new polices, strategies and interventions that are designed and target to enable every woman hoped to help women fully decide by themselves. 

Women alone family planning use decision making increased significantly from 2014 to 2020. This 20% change in the level of women alone decision making is promising and in line with national and international indicators, targets, and strategies. However, it is important to continue monitoring progress, addressing barriers, and ensuring sustained efforts to achieve the desired goals. 

The findings of substantial regional disparities in women's alone contraception decision-making highlight the need for targeted interventions in regions with lower rates. Some potential strategies to address these disparities could include setting up regional-specific intervention such as tailoring family planning programs to address the unique cultural and social factors in each region, ensuring that women have access to accurate information and resources to address misconceptions, reduce stigma, and emphasizing the importance of women's participation in decision-making. 

This study identified factors which is positively and negatively related with women alone and joint decision making on family planning use. Activities and efforts that help women to be able to obtain the method she desired to use, improve her physiological readiness and trust on her own to use family planning methods contribute their share in improving women alone decision making. Additionally, the findings calls up on creating awareness on reproductive health service availability and accessibility including maximizing use of youth friendly service unit thereby empowering young/adolescent’s and girls to use family planning method.

Activities and intervention that improve husband and/or partner involvement in his wife´s family planning use decision making process such as improving his feeling on his wife family planning use, balanced discussion on family planning use where the women say is equally entertained before using the method and encourage women to decide to use family planning via religious leaders mainly Muslim religious leaders, informing the possible side effects and what to do if the women encounter serious side effect during her family planning use visits are hoped to improve women alone and joint family planning use decision making by the couples. 

Finally including missing variable such as husband desired number of children, age difference, husband education and employment, women employment for future research in general and performance monitoring for action (PMA) survey in particular.”

**Detailed comments from Reviewer 1** 

The following comment was raised by reviewer 1 on date 8 Aug 2023 and the authors provided response under the point by point response during the first revision submission. 

Comment 1: make clear your study which type of family planning (long term or temporary family planning)

Response: The analytic sample is restricted to women who have been using or most recently used modern contraceptive methods which is also comprise both long term or temporary family planning methods. Assuming the term long term is to mean permanent method and long-acting reversible methods, our study includes Male and Female sterilization, IUD and Implant.

Those modern temporary family planning methods also included such as injectable, pills, emergency pills, male and female condom. 

Comment 2: according to PMA there are some additional factors involved but not discussed in your study.

Response: This study tried to exhaustively include as many as 20 variables in the model which are summarized in the descriptive section. Selection of important confounders was based on available empirical evidence in the area of FP use decision making.

* The methods are generally appropriate, although clarification of a few details and provision of a rationale for the use of this particular method of measuring Women Decision Making on Use of Modern Family Planning Methods and Associated Factors, Evidence from PMA Ethiopia.

Response: We appreciate this concern, and the following two clarifications are mentioned. To start with, detail clarification about PMA data is also available in the method sample size and selection technique part in the main manuscript. The following paragraph is also added in order to clarify the rationality to use PMA data in the same section. 

The rationality to use PMA data includes that currently, PMA data is the best available recent and real time data even used by the minster. In addition, PMA collects data by resident enumerators using smart phone with customized ODK application which facilitates real time data collection and timely feedback in correcting errors.

If the concern is mean to the measurement of the outcome variable, family planning decision making, as can be witnessed form the listed studies in the reference and clearly justified in the introduction part ,the absence of standardized measurement is a problem; most studies which include further analysis of Demographic and Health Survey (DHS) data using a single question to measure FP use decision making while few fragmented studies try to used inconsistent constructs to create a composite variable without the evidence on the constructs reliability and reducibility. 

The following references have been added to the manuscript Discussion part. 

1. UN. Transforming Our World: The 2030 Agenda for Sustainable Development. New York; 2015.

2. UNFPA. GLOBAL GOALS INDICATOR 5.6.1: Research on factors that determine women’s ability to make decisions about sexual and reproductive health and rights. Volume I, 2019.

3. Stover J, Sonneveldt E. Progress toward the Goals of FP2020. Studies in family planning. 2017;48(1):83-8.

4. Brown W, Druce N, Bunting J, Radloff S, Koroma D, Gupta S, et al. Developing the “120 by 20” goal for the Global FP2020 Initiative. 2014;45(1):73-84.

5. FMoH. RH Strategic Plan - Ethiopia 2021-2025. 2021.

6. FMoH. Health Sector Transformation Plan II (HSTP-II). 2021.

END

---

## [Editor Report · Decision Letter 2]

26 Jan 2024

Women Decision Making on Use of Modern Family Planning Methods and Associated Factors, Evidence from PMA Ethiopia

PONE-D-23-04330R2

Dear Dr. FANTAYE,

We’re pleased to inform you that your manuscript has been judged scientifically suitable for publication and will be formally accepted for publication once it meets all outstanding technical requirements.

Within one week, you’ll receive an e-mail detailing the required amendments. When these have been addressed, you’ll receive a formal acceptance letter, and your manuscript will be scheduled for publication.

Kind regards,

Takele Gezahegn Demie, MPH

Academic Editor

PLOS ONE

Additional Editor Comments (optional):

Dear Author,

Thank you for submitting your manuscript to PLOS ONE. After careful consideration, I feel that it could be acceptable for publication.

However, I suggest in-house checking to further point out whether it is consistent with the PLOS ONE submission guideline and publication.

I hope PLOS ONE editorial staff will communicate with you regarding further queries if any.

Good LUCK
---

## [Editor Report · Acceptance letter]

8 Feb 2024

PONE-D-23-04330R2 

PLOS ONE

Dear Dr. Fantaye, 

I'm pleased to inform you that your manuscript has been deemed suitable for publication in PLOS ONE. Congratulations! Your manuscript is now being handed over to our production team.

Kind regards, 

on behalf of

Mr. Takele Gezahegn Demie 

Academic Editor

PLOS ONE